# Effects of Various Mineral Admixtures and Fibrillated Polypropylene Fibers on the Properties of Engineered Cementitious Composite (ECC) Based Mortars

**DOI:** 10.3390/ma15082880

**Published:** 2022-04-14

**Authors:** Muhammad Hanif Khan, Han Zhu, Muhammad Ali Sikandar, Bakht Zamin, Mahmood Ahmad, Mohanad Muayad Sabri Sabri

**Affiliations:** 1Department of Civil Engineering, Tianjin University, Tianjin 300350, China; hanifkhan_88@tju.edu.cn; 2Key Laboratory of Coast Structure Safety of the Ministry of Education, Tianjin University, Tianjin 300350, China; 3Department of Civil Engineering, CECOS University of IT & Emerging Sciences, Peshawar 25000, Pakistan; bakht@cecos.edu.pk; 4Department of Civil Engineering, Faculty of Engineering, International Islamic University Malaysia, Jalan Gombak, Selangor 50728, Malaysia; ahmadm@iium.edu.my; 5Department of Civil Engineering, University of Engineering and Technology Peshawar (Bannu Campus), Bannu 28100, Pakistan; 6Peter the Great St. Petersburg Polytechnic University, 195251 St. Petersburg, Russia

**Keywords:** durability, mechanical performance, materials characterization, microstructure, polypropylene (PP) fibers, UPV

## Abstract

This study investigates the mechanical and durability properties of fly ash-based engineered cementitious composites (ECC). The effect of various mineral additions, such as wheat husk ash (WHA), rice husk ash (RHA), glass powder (GP), and fibrillated polypropylene (PP) fibers, on mechanical performance, water absorption, and porosity was investigated. Furthermore, the durability of ECC specimens was assessed in terms of sorptivity, acid/sulfate attacks, electric resistivity (ER), rapid chloride penetration (RCPT), and ultrasonic pulse velocity (UPV). The results revealed higher mechanical strength, UPV, and ER values for RHA-based ECC. After 180 days of immersion in acid and sulfate solutions, RHA-based ECC showed a lower loss in compressive strength (23.21% and 1.07% in HCl and Na_2_SO_4_, respectively) relative to the control mix (44% and 7% in HCl and Na_2_SO_4_, respectively). Moreover, analytical characterizations such as X-ray diffraction (XRD), Fourier transform infrared (FTIR), Scanning Electron Microscopy (SEM), and Energy dispersive X-ray (EDX) analyses were also carried out to corroborate the mechanical and durability properties of ECC.

## 1. Introduction

Durability, ductility, and sustainability of ordinary concrete can be increased with specially designed high-performance fiber-reinforced cementitious composites (HPFRCC) types called engineered cementitious composites (ECC). Higher mechanical strength can be achieved by using a high volume of fibers used in HPFRCC [1]. ECC is a ductile and bendable type of concrete containing reinforced polymer fibers with a strain hardening capacity in the range of 3–7% [2]. The high ductility of ECC is due to the combination of fiber-matrix interfaces in the paste [3]. Li et al. [4], reported that the tensile strain hardening properties of ECC were about 500 times that of normal concrete and fiber reinforced concrete (FRC). It is reported that ECC can control the crack width in concrete [5,6,7]. The self-healing properties improve the tensile strength and chemical durability of ECC. Essential constituents of ECC may include cement, fly ash, fine quartz sand, polymeric filament fibers, water, and superplasticizer [1].

The high volume of fly ash produces relatively ductile ECC [8]. Moreover, the high volume of fly ash can improve better crack width control and minimize drying shrinkage, which is good for long-term serviceability [8]. ECC containing fly ash can withstand different weathering conditions, corrosion resistance, freeze-thaw cycles, water permeability mechanisms in cracked concrete [1] and cyclic loading conditions. The fly ash, a pozzolanic material, improves the mechanical properties of ECC, promotes secondary hydration in concrete and cement, and modifies the concrete microstructure, leading to the self-healing property of the concrete [8,9,10]. The fly ash is obtained from power plants by combusting coal and is used worldwide. Besides fly ash, there have been studies about other sustainable mineral additions to ECC that also enhanced its durability [11]. Regarding environmental protection, it is a imperative to utilize waste materials in cement concrete. The construction industry utilizes various mineral admixtures to replace cement to improve concrete characteristics. These admixtures commonly include fly ash, silica fume, rice husk ash (RHA), wheat husk ash (WHA), slag, metakaolin, etc. Sahmaran et al. [12] and Li et al. [13] studied the durability and tensile strength of ECC mixture in a chloride environment. They observed better durability properties and higher tensile flexibility. Engineered cementitious composites utilizing RHA demonstrated enhanced durability properties by improving capillary resistance and capillary pore tortuosity; moreover, the use of RHA in ECC mix minimizes the risk of free and restrained shrinkage [1]. The resistance against chloride penetration and acid attack can be improved with the RHA in mortar; additionally, the result indicates increased compressive strength and decreased permeability [14]. Experiments show that RHA used in mortars results in high ductility, resistance to crack formation, lower voids content, water absorption, and heat of hydration [15]. Wheat Husk Ash used in the replacement of cement can increase mechanical and physical properties, and accelerate early age hydration reactions; later age calcium hydroxide content is reduced and due to finer particles, gives a compact microstructure [16]. The negative impact of PP fibers in mortar can be eliminated due to fine particles of wheat straw ash [14]. The study shows that the electric resistivity, ultrasonic pulse velocity, and compressive strength of GP mortar can improve with fly ash inclusion or granulated blast furnace slag as binary binders. Additionally, resistance to sulfuric acid attack was enhanced when GP was used as up to 45% of cement replacement [17]. Recycled glass powder (RGP) also improves durability and mechanical strength [18]. Said et al. [19] studied RGP used in HVFA-ECC (high volume fly ash engineered cementitious composite), FA (fly ash) to cement ratio of 1.2, improves mechanical performance, electrical resistivity, chloride ion resistance results, imparting ductility to standard ECC. Said et al. [17,19] confirmed that using recycled waste glass powder enhances the chemical resistance of concrete and mechanical properties of ECC. It is stated elsewhere that incorporating glass particles of a larger size (~3 mm) into the concrete can make it vulnerable to alkali–silica reaction (ASR), leading to cracking and expansion. Nevertheless, the finer glass particles lower the probability of alkali–silica aggregate reaction and expansion up to 50%—these has been extensively experimented with, along with fly ash [20,21,22].

Fibers are used in concrete for various construction applications, due to their greater performance than ordinary concrete. Different fibers, such as polyvinyl alcohol (PVA), polypropylene (PP), steel, plastic, carbon, glass, etc., are used in mortar and concrete. The dispersion of fibers in the concrete microstructure can improve the performance against plastic shrinkage and chloride ion penetration. [23]. According to Latifi et al. [24], PP fibers’ utilization is cost-effective and improves durability properties. PVA (polyvinyl alcohol) is one of the most commonly used fibers in concrete, having high stiffness and high tensile strength capable of developing stronger interfacial solid bond [25]. Long-term ductility can be maintained with ECC containing polyvinyl alcohol (PVA), recommended for a wide range of structural applications [26]. Experiments indicate that PP fibers are more viable than PVA fibers; nonetheless, both types of fibers show similar mechanical performances. ECC fabricated with RHA and PP fibers are greener materials that show an average strain hardening ability of 1.2% and flexural and tensile strength up to 6% higher than control [1,23]. It is well known that incorporating PP in ECC can improve the toughness, ductility, dissipated energy, and impact resistance of concrete materials [27]. Karahan and Atis [28], studied PP fibers that can reduce workability and drying shrinkage and increase freeze-thaw resistance and compressive strength.

Polypropylene fibers positively impact concrete’s physical, mechanical, and thermal qualities. Furthermore, PP fibers enhanced water absorption, porosity, and capillary resistance as the fiber content increased. These demerits can be eliminated using different mineral admixtures by densifying the microstructure [28]. According to research, fine elements such as fly ash are essential for the uniform dispersion of fibers in the cementitious system [24].

The prime aim of this study was to investigate the synergistic effect of industrial and agricultural waste such as FA, RHA, WHA, GP, and Fibrillated Polypropylene (PP) fibers in controlling the mechanical and durability properties of ECC mortars. According to the Paris agreement [21], fly ash production is limited; hence the search for other mineral admixtures is imperative. RHA, WHA, and GP are sustainable, cheaper, and locally available waste minerals, which can be used instead of cement and fly ash. For this study, the choice of PP fibers was due to their low density, high strength, ease of use, low cost, and higher chemical resistance. In addition, the characterization of the microstructure of ECC has been evaluated. This study concludes that using industrial and agro-waste based mineral admixtures in ECC can offer various benefits to the construction industry, such as cost-effective solutions, including low environmental impact and improved construction efficiency.

## 2. Materials and Methods

### 2.1. Materials

Ordinary Portland cement (OPC) Type-1, having a specific gravity of 3.10, purchased from Kohat cement factory conforming to ASTM C150 [29] was used. Table 1 collects the oxide composition of cement and of all mineral admixtures. The fly ash (FA), rice husk ash (RHA), wheat husk ash (WHA), and glass powder (GP) used had a specific gravity of 2.20, 2.20, 2.04, and 2.70, respectively. The river sand complied with ASTM C128 [30], with a specific gravity of 2.62, and fineness modulus of 2.68 was used.

Class F fly ash conforming to ASTM C618 [31]. The raw wheat husk was obtained from a local agriculture field in Peshawar, Pakistan, converted into ash at 650 °C by the method adopted in the literature [14]. The difference in SiO_2_ content used in other studies varies in the soil and environmental conditions. Fly ash, high-range water reducer (HRWA), and synthetic fibrillated polypropylene (PP) fibers were collected from Sika chemicals, Pakistan. Rice husk ash and glass powder were purchased from PCSIR laboratories, Pakistan. Figure 1 presents the gradation curve for various constituents of ECC mixes. Table 2 gives the properties of PP fibers, while Figure 2 presents the visual aspect of mineral admixtures and PP fibers used.

### 2.2. Mix Design

The study was carried out on eight designated ECC mixes. All mortar mixtures contain OPC and FA as primary binding materials. While only OPC and FA were used in the control mixtures (C). Whereas the remaining mixtures were fabricated, replacing portions of FA with RHA, WHA, and GP. Water reducing admixture or superplasticizer is used in ECC to maintain the workability and slump properties of the paste because the inclusion of PP fibers can reduce the workability of the mortar significantly. ECC mixes contained the same amount of cement, the same w/cm of 0.25, and the same amount of HRWRA (high range water reducer admixture), i.e., 1.2% by the weight of the binder. For all ECC mixtures, mineral admixtures, fibers, OPC, and sand were first thoroughly mixed, followed by the addition of the required amount of water and HRWRA. All mixtures were prepared with a power-driven standard Hobart pan mixer, following ASTM C192 guidelines [32]. The mix proportions for ECC mixes are collected in Table 3. Ordinary Portland cement, sand, mineral admixtures, and fibers were blended for about sixty seconds in a power-driven mixer. After that, HRWRA that had been mixed in water for four minutes was poured into the mixer. The mixture was cast and tamped into different molds for mechanical strength and durability properties. Mortar samples were compacted into three layers using a temping rod. For 24 h, the specimens were covered in plastic sheets. Subsequently, the specimens were extracted from the molds and placed in the curing tank at room temperature and relative humidity of 90 ± 5%. After completing the curing period, various experiments were performed on ECC specimens.

### 2.3. Methods

#### 2.3.1. Mechanical Strength

Mechanical properties of ECC specimens were assessed in terms of compressive and flexural strength, in accordance with ASTM C109 [33] and ASTM C348 [34], respectively. The compressive strength of a 50 mm side cube sample was measured at 7, 28, 56, and 90 days. Flexural strength testing on 28 day- water cured 40 × 40 × 160 mm^3^ prism samples was performed. Three samples were tested for each mix proportion and the average result is reported in this study.

#### 2.3.2. Water Absorption and Porosity

Mortar samples from each mix proportion were weighed, water cured for 28 days at room temperature (20 °C), surface dried, and reweighted to assess the amount of water absorbed and the changes in porosity, as per the method given in ASTM C642 [35]. Three samples for each mix proportion were tested to reach an average value.

#### 2.3.3. Sorptivtity

The sorptivity measures the water absorption rate by capillary suction of the mortar samples. The sorptivity after 28 days of cured ECC cube specimens was measured according to ASTM C1585 [36]. A sealable layer was used to protect all specimens surfaces or taps except the surface exposed to water for water penetration measurement through the exterior mortar surface. The specimens’ mass was noted every 30 min for 2 h consecutively after putting them in a water chamber at an immersion depth of 5 cm. Three samples from each mix proportion were tested. The sorptivity was determined using the following formula.
(1)$$S=\frac{I}{t^{0.5}}$$

A straight line is drawn when the amount of water is plotted against the square root of the exposure period. Here, I (mm) indicates the total water absorbed, S is the sorptivity coefficient, and t (√h) is the sample’s water exposure period.

#### 2.3.4. Ultrasonic Pulse Velocity (UPV), Electric Resistivity (ER), Rapid Chloride Penetration Test (RCPT)

Specimens that had been hydrated for 28 days were evaluated for UPV, following ASTM C597 guidelines [37]. The same ECC specimens were tested for electrical resistivity (ER), using an ER measuring device with two electrodes. The ER test method used on mortars was reported by Tran et al. [38]. In this test, the electrodes were placed on the specimen on both sides, and ER was measured using the following equation:(2)$$\mathrm{ER}=\frac{\mathrm{RA}}{L}$$
where R (kΩ) denotes the resistance, A (cm^2^) is the area of cross-section, and L (cm) is the length of the sample.

The RCPT test was performed on a water-soaked, 50 × 100 mm thick diameter ECC specimen following ASTM C1202 [39]. In this test, ECC specimens were subjected to a DC (Direct Current) voltage of 60 V for 6 h, where one side of the sample was in contact with a 3.0% NaCl solution and the other side was immersed in contact with a 0.3 M NaOH solution. The total charge passed was measured in coulombs.

#### 2.3.5. Acid and Sulfate Resistance

Acid and sulfate resistance tests were performed according to ASTM C267 [40]. After curing for 28 days and drying at 105 °C in an electric oven, the ECC cube samples were placed in two separate water tanks containing 5% hydrochloric acid (HCl) and 5% sodium sulfate (Na_2_SO_4_) solution. The solution medium was renewed every 30 days. The specimens were kept in a plexiglass tray with a cover to avoid evaporation and keep the PH constant in the medium. Before placing samples in acid and sulfate solution, the specimen’s dry weight was recorded. After every 30-day interval for 180 days, the samples were taken out of the acid and sulfate solution tanks to determine their dry weight and retained compressive strength. Three samples from each mix proportion were examined.

#### 2.3.6. Materials Characterization

Fourier transform infrared (FTIR, Nexus 870, Thermo Nicolet Corps, Waltham, MA, USA) spectroscopy, X-ray diffraction analysis (XRD, Ultima 111, Rigaku Inc., Tokyo, Japan), scanning electron microscopy (SEM, JSM 5910 JEOL, Tokyo, Japan), and energy dispersive X-ray spectroscopy (EDX, INCA 2000, Oxford Instruments, High Wycombe, UK) were employed to assess the hydration characteristics of 28 day cured ECC specimens.

## 3. Results and Discussion

### 3.1. Mechanical Strength

Figure 3a shows the compressive strength for all ECC mixtures with increasing curing time. The compressive strength of the R20, R10, W20, W10, CMB, GP20, GP10, and C is 47, 42, 38, 33, 29, 26, 24, and 23 MPa, respectively, after 28 days, which increased to 56, 53, 48, 42, 39, 37, 34, 30 MPa, respectively, after 90 days of curing. Further, as can be seen, RHA and WHA-based mortars gain higher strength than other mortars. Both pozzolanic reaction and the micro filler effect contributed to RHA and WHA-based ECC strength gain compared to control mortar [41,42]. The micro-filler effect of RHAs distributes hydration products more homogeneously in the available space, resulting in a much denser matrix [43]. RHA particles form an additional CSH gel through secondary hydration, resulting in a more compact and dense microstructure [44]. On the other hand, waste glass powder (GP) primarily induces the filler effect. After 7–10 days, its hydration liberates enough lime to initiate the secondary pozzolanic reaction. As a result of this reaction, a larger amount of calcium-silicate hydrate (C-S-H) is produced. The use of GP in concrete benefits continued strength gain over time due to pozzolanic action [45]. RHA is mainly made up of amorphous silica (85–90%) and has a microporous structure that makes it ideal for use as a cement substitute due to its pozzolanic reaction. The amorphous silica of RHAs can react with calcium hydroxide crystals that form during concrete hydration, and secondary C-S-H gel formation can fill up the concrete pore structure. RHA may also fill in gaps between cement particles and reinforce the interlocking of concrete mixtures [46]. As a result, a denser matrix with increased strength can be achieved. The lower compressive strength of ECC mixtures containing GP compared to RHA and WHA is the low reactivity of GP, which results in paste toughness [47,48]. The inclusion of fibers in the paste can improve the compressive strength due to reducing crack formation development [49].

Figure 3b represents the flexural strength of various ECC specimens. In this figure, flexural strength is observed increasing for ECC specimens in which fly ash was replaced by the RHA, WHA, and GP. It is known that the ECC failure mechanism is different from that of ordinary concrete; ECC does not fail on the first crack because the fibers used in ECC can counter the pull-out effect after the first crack [19]. The flexure strength of R20 was 6.5 MPa, while for the control mixture the flexure strength was 5.1 MPa. An enhancement in flexural strength was noticed due to PP fibers. The crack-bridging properties of the PP fibers led to an increase in the flexure properties of ECC.

Furthermore, the aspect ratio and length of the fibers can also affect flexural strength of ECC. Longer fibers (with a higher aspect ratio) are more effective at bridging cracks than shorter fibers. The addition of PP fibers is responsible for the increased flexural strength, which improves concrete ductility. This is due to a fiber concrete composite behavior, which prevents the fiber concrete element’s brittle failure. A stronger frictional bond for the PP fiber system with cementitious microstructure may boost composite ductility and strain hardening capacity [25].

### 3.2. Water Absorption and Apparent Porosity

Water absorption and porosity results are presented in Figure 4. Figure 4a,b shows that as the content of mineral admixtures in the ECC specimen increases, the water absorption and the apparent porosity decrease. Figure 4a shows that water absorption for R20, R10, W20, W10, CMB, GP20, GP10 and C for 28 days is 3.75, 3.85, 4.1, 4.28, 4.8, 4.36, 4.58, and 5.4%, respectively. While the apparent porosity for the same was 11.1, 11.2, 11.8, 12, 14.1, 12.92, 13.2, and 14.9%, respectively. RHA shows lower water absorption and porosity levels, followed by WHA, CMB and GP. RHA and other mineral admixture particles filled large voids and refinement of microstructure [44,50]. Righi et al. [15] found that RHA can reduce water absorption and voids in the paste. As GP is added to the mixture, it increases absorption compared to other mineral admixtures. The slow reactivity of GP results in higher porosity and water absorption of cement mortars [51]. They noticed an increase in absorption as the amount of glass powder used to replace cement in concrete mixtures increased. These findings show that increasing the porosity of ECC mixtures containing GP increases the number of pathways for water to enter the composite. All formulations containing RHA, WHA, and GP showed reduced water absorption compared to the control. This is due to the pozzolanic reaction between silicon dioxide and calcium hydroxide in hydrated cement paste, leading to CSH formation. In fact, the pozzolans effectively lower the system’s porosity, thereby increasing the durability of cementitious materials [52]. The anti-cracking ability of PP fibers [49] reduces permeability, ensuring water tightness.

### 3.3. Sorptivity

The rate of water uptake in the ECC specimens, which decreases with the increase in the dosage of mineral admixtures, is given in Figure 4c. Among all ECC specimens, R20 mix specimens indicated the lower tendency of capillary rise, followed by WHA, CMB, GP, and C. Generally, mineral admixtures are used as filler in ECC to densify the composite’s microstructure at an initial stage. Matos and Coutinho [53], found that the sorptivity of mortar mixtures containing GP was not considerably different from that of the control. Zhang [54], noted that PP fibers used with fly ash and silica fumes could significantly reduce the water permeability of composites. This is because PP fibers’ anti-cracking capacity reduces the micro-cracks in the concrete and the water pathway entering the concrete.

### 3.4. Measurement of Weight Loss

The reaction of hydration products such as portlandite and CSH with monoprotic acid (HCl) is expressed by the equations mentioned below (Equations (3) and (4)).
Ca(OH)_2_ + 2HA → Ca^2+^ + 2A^−^ + 2H_2_O(3)
xCaO.ySiO_2_.nH_2_O + 2xHA → xCa^2+^ + 2xA^−^ + ySi(OH)_4_ + (x + n − 2y) H_2_O(4)

The formation of calcium chloride is due to the consumption of hydration products in HCl, as determined by Equations (3) and (4). The salts mentioned in the above equations result in the weight loss of ECC mix proportions due to their high solubility. All ECC specimens underwent weight loss during the immersion period in acidic media and their rate of weight loss increased with an increase in immersion time. As shown in Figure 5a, the weight losses in the acidic media for R20, R10, W20, W10, CMB, GP20, GP10, and C mix specimens are measured to be 4.1, 4.5, 4.8, 5.23, 5.8, 5.99, 6.31, and 6.82%, respectively, as shown in Figure 5a. The R20 mix specimen underwent a lesser weight loss than other specimens, which shows its better performance in the acidic medium.

Since more soluble calcium salts are formed due to acid attack on ECC specimens, while less soluble gypsum precipitation occurs due to sulfate attack, an oxide mineral (portlandite) acquires a solution. In sulfate attack on ECC specimens, a hydrous calcium aluminum sulfate mineral (ettringite) is formed. Moreover, secondary gypsum is formed at higher concentrations (1–2 g/L) of sulfate. The given Equation (5) is used in sulfate attack on ECC specimens.
Ca(OH)_2_ + Na_2_SO_4_ + H_2_O → CaSO_4_·2H_2_O + 2Na^2+^ + 2OH^−^(5)

Similarly, ECC specimens also underwent weight change in the sulfate solution as shown in Figure 5b. However, their weight loss rate differed from those in acidic media. The gain in weight of the specimens was noticed for up to 90 days. After 180 days of immersion in the sulfate solution, the weight loss for all specimens was noted. The weight loss was lower in the sulfate compared to acidic solution due to the leaching of portlandite. The gain in weight after up to 90 days is due to the utilization of sulfate ions into the cement matrix, while the loss in weight is due to the expansion mechanism of gypsum after 90 days [55]. The formation of expansive ettringite in the control specimen results in a weight loss of the sulfate solution. RHA significantly reduces the permeability of the concrete, which results in high resistance to HCl solution. Ramasamy [43] reported that RHA used in concrete leads to high acid and sulfate resistance.

### 3.5. Changes in Compressive Strength after Acid and Sulfate Attack

ECC specimens subjected to acidic and sulfate attack for 180 days was analyzed for residual compressive strength and their results are shown in Figure 6a,b.

The residual compressive strength of ECC mixes, i.e., R20, R10, W20, W10, CMB, GP20, GP10, and C, after 180 days of immersion in HCl were 43, 40, 35, 29.2, 26, 25.6, 22.8, 16.8 MPa, respectively. In all these mixes, the R20 mix can be regarded as more resistive to acidic attack, showing a lesser decrease in compressive strength than other mixes relative to C mix specimens. The acid depletes the portlandite in the control ECC mix in the acidic media blister the surfaces of the other ECC specimens. Similarly, R20, R10, W20, W10, CMB, GP20, GP10, and C mixes showed a compressive strength of 55.4, 52, 47, 41, 38, 35, 32, and 27.9 MPa, respectively, as cured in a sulfate solution up to 180 days. In sulfate solution, the gain in compressive strength was observed for up to 90 days. After that, the compressive strength gradually decreases. The increase in strength is due to the sulfate reaction with the hydration products resulting in the formation of gypsum and ettringite [55].

### 3.6. Ultrasonic Pulse Velocity (UPV) Test

The ultrasonic pulse velocity (UPV) test is used in concrete materials to assess cracks, defects, and delaminations. Figure 7 depicts the ultrasonic pulse velocity for all ECC mixtures at 28 days. R20 exhibited the highest pulse velocity, followed by R10, W20, W10, CMB, GP20, GP10, and C mix specimens. R20, R10, W20, W10, CMB, GP20, GP10, and C mix specimens pulse velocity values were 5200, 4950, 4870, 4800, 4700, 4650, 4610, 4410 m/s, respectively, after a period of 28 days. The dense and solid microstructure formation with lesser voids in RHA and WHA mixtures results in higher UPV values. The porosity of mortar directly relates to UPV because discontinuities in the pores delay wave propagation [56]. The inclusion of fine-sized WHA particles also enhanced the UPV value [16]. The increment was much higher for finer ash particles. Material properties, mix proportion, pore structure, and the interfacial zone between aggregates and cement paste are well-known factors influencing the UPV values. It has been observed that ECC-based mortars have improved UPV results over traditional mortars because of their higher paste volume.

As previously mentioned, the addition of RHA and WHA refines the pore structure. At the same time, the C-S-H gel fills the capillary pores. In RHA and WHA based ECC, the capillary pores are reduced by the accumulation of calcium-silicate hydrate (C-S-H). CSH segments the pore structure, resulting in more homogeneous and impervious concrete. Due to the phenomenon mentioned above, the pores in the cement matrix are reduced, leading to delayed propagation of the UPV values [57].

### 3.7. Electrical Resistivity (ER)

Electrical resistivity (ER) is a useful method for testing the incidence of corrosion of steel in concrete materials. The electrical resistivity test results for all ECC-based mortars are presented in Figure 8. Their beneficial effects are even more apparent at 20% substitution of RHA and WHA. ER values for R20, R10, W20, W10, CMB, GP20, GP10, and C mix specimens were 21, 20, 19.5, 18.9, 17.9, 17, 16, 14 kΩ-cm, respectively, after a period of 28 days. Compared to other mortars, RHA and WHA show higher ER values at 28 days. The percent improvement in ER at 28 days for R20, R10, W20, W10, CMB, GP20, and GP10 compared to C is observed to be 33.33%, 30%, 28.20%, 25.92%, 21.788%, 17.64%, and 12.5%, respectively. This improvement is due to the ability of pozzolan silicates to react with alkalis, which in turn reduces the concentration of OH-ions in mortar systems [57]. The pozzolans also help reduce the ionic concentration of pore solution ions, leading to and preventing ionic diffusion. Additional CSH is formed when SiO_2_ of RHA, WHA, and GP combines with calcium hydroxide. The microstructure of mortars is affected by this reaction, which reduces the interpore connectivity, resulting in a dense and compact microstructure [57]. All ECC mixes met the low corrosion probability criteria outlined in ASTM guidelines. This is probably due to the formation of compact and dense microstructures due to the higher surface areas of WHA, RHA, and GP. In addition, PP fibers can also improve ER value of concrete and reduce rebar corrosion probability [49].

### 3.8. Rapid Chloride Penetration Test (RCPT)

The total load passed in 6 h was reported as a chloride permeability measure in this test. Figure 9 shows the RCPT values of ECC mortars hydrated for 28 days in terms of charge passed. The same figure shows that the RCPT values were 450–920 °C at 28 days for the control mixture and a mixture containing RHA, WHA, and GP, which characterized low chloride permeability following ASTM C1202. Furthermore, the introduction of RHA, WHA, and GP substantially reduced the charge on mortars. In addition, the permeability of chloride ions is reduced by increasing the replacement rate of RHA, WHA, and GP. The RCPT values for R20, R10, W20, W10, CMB, GP20, and GP10 were lowered by 51.08%, 48.69%, 44.56%, 41.30%, 35.86%, 22.82%, and 14.13%, respectively, concerning control specimen after a period of 28 days. The decrease in RCPT values is due to microstructure densification and reduced pore interconnectivity of ECC. It is presumed that these mineral admixtures with minimal size have been able to produce a packed and compact microstructure, which offered better resistance to chloride penetration. Based on the ASTM 1202 criterion, RHA, WHA, and GP mixtures are graded as very low chloride permeability mortars. The higher charge passed for ECC mixtures containing GP is due to the GP particles’ slower pozzolanic reactivity and higher porosity than RHA and WHA [47]. The addition of GP increases the porosity of the composite, making it easier for chloride ions to pass through it. A previous study shows that increasing the content of GP in ECC can increase the total charge passed [19]. Schwarz and Neithalath [48] observed lower chloride permeation for concrete mixtures containing glass powder than the control mix.

The RCPT results were consistent with physical and mechanical property test results. The control mixture also demonstrated the resistance to chloride ion permeability, having only ordinary Portland cement and fly ash. Ozbay et al. [58], replaced cement with mineral admixtures, which reduced ion concentrations and microstructure densification. The chloride ions in the cement matrix diffuse through permeable voids. Chloride permeability is also influenced by the chloride binding capacity of constituents of cementitious mix. Some chlorides may be combined with tri-calcium aluminate (C_3_A) to form stable chlorocomplexes. The free chloride species stay in the mixture and allows initiating corrosion. Additionally, FA helps reduce the RCPT values due to its high alumina content. Major factors influencing the RCPT values are total porosity, size, continuity of voids, permeability, and diffusivity. The polypropylene fibers reduce permeability by preventing the propagation of tiny fractures in the concrete [24]. Figure 10 represents the testing setup and schematic diagrams for UPV, ER, and chloride penetration tests.

### 3.9. Fourier Transform Infrared Spectroscopy Analysis (FTIR)

FTIR analysis was employed for identifying distinct functional groups in ECC mixes. Figure 11 depicts the FTIR transmission spectra of 28-day hydrated R20, W20, CMB, GP20, and control samples. Each FTIR spectrum showed major bands at 3637, 1640, 1411, 1120, 960, and 970 cm^−1^, for O–H stretching of portlandite, υ_1_ and υ_2_ stretching of molecules of water, υ_2_ bending of the water molecule, υ_3_ stretching of carbonate, Si-O stretching of silicate (C-S-H), υ_2_ stretching of carbonate, and υ_4_ stretching of carbonate, respectively [55,59]. Bands at 3427 cm^−1^ to 3636 cm^−1^ indicate chemisorbed water and surface OH groups. The bands between 1632 cm^−1^ and 1647 cm^−1^ are the C-O group. A double peak characterizes the C_2_S and C_3_S phases at 995–900 cm^−1^ and 938–883 cm^−1^, respectively, assigned to Si-O asymmetric stretching vibrations [60]. The band at 960–1120 cm^−1^ indicates the formation of CSH gel [61]. The same figure shows that R20 and W20 have more CSH development than other minerals, so R20 and W20 can resist acid and sulfate attaching and long-term durability than other mineral additions. IR peaks show that all mineral admixtures behave well due to their pozzolanic reaction and micro-filler effect. According to Habeeb et al. [62] and Hamid et al. [63], finer pozzolans help improve the strength of ECC specimen by pozzolanic reaction with CH, resulting in the formation of CSH gel.

### 3.10. X-ray Diffraction Analysis (XRD)

Figure 12 shows the XRD patterns in the 2θ range for ECC specimens to detect the crystalline hydration products. Ettringite, hexagonal prismatic crystals of calcium hydroxide, quartz, illite, belite, calcite, and C-S-H were detected in the XRD patterns of all mix formulations.

In all the XRD patterns, intense peaks due to portlandite can be identified at 2θ values of 18.09°, 34.09°, 47.12°, 50.79°, 54.34°, 59.30°, 62.54°, and 64.23° [64,65,66]. The intense peaks at 2θ values of 20.85°, 26.65°, 36.54°, 39.46°, 50.14°, 59.95°, and 67.75° are due to the presence of quartz (Q) [67]. Although C-S-H is difficult to detect on XRD patterns due to its amorphous nature, its presence can be identified by peaks appearing at 2θ values of 31.92° and 49.76° [68]. X-ray diffraction peaks at 2θ 29.4°, 32.22°, and 32.66° appeared due to alite (a) phase and belite (b) phase [59]. The formation of calcite (c) can be detected at 2θ values of 29.33°, 36.1°, 38.19°, 43.5°, and 57° [17]. Additionally, the depletion of portlandite occurred due to the portlandite conversion into the CSH phase. In the control ECC, gypsum (G) formation is observed at 2θ values of 37.5° [17]. The conversion of calcium hydroxide into CSH in these mixes increases compressive and flexural strength. Moreover, resistance to acid and sulfate attacks was also enhanced. [52]. XRD patterns confirmed the presence of the C-S-H phase at 31.92° and 49.76°, which enhanced mechanical properties.

### 3.11. Microstructural Properties

Scanning electron microscopy (SEM) and energy dispersive X-ray analysis (EDX) analyses were performed to determine the effect of mineral admixture on the micro structural properties of ECC mixtures. The EDX and SEM analyses of ECC mixtures are shown in Figure 13a–e. Microstructure study reveals that mineral additives disperse the fibers uniformly in the matrix [24]. Uniform dispersion of fibers haves the ability of crack bridging, reducing the cracks and imparting ductility in the system.

Mineral admixtures can help form the dense and compact microstructure with additional CSH in ECC mixtures compared to the control. This is probably due to the fine particles of mineral admixtures that can fill the pores in the paste, leading to a dense microstructure with greater strength. Figure 13a shows that RHA has a more compact microstructure because RHA fine particles also act as a filler to reduce the pores, reducing the system permeability, sorptivity, and porosity. As a result, moisture and chloride ions do not enter the cement matrix, increasing durability. WHA and GP act the same as RHA but to a lesser extent. Due to its pozzolanic nature, these mineral admixtures in ECC samples can accelerate the hydration reaction. As a result, more hydration products are formed. This explains why ECC mixtures underwent lower weight losses in acid and sulfate environments.

The control ECC sample in Figure 13e shows many interrupted pores with the lesser amounts of CSH gel. The weight % of elements based on the EDX analysis of ECC mixtures is listed in Table 4. Ca/Si ratio is for R20, W20, CMB, GP20, and C is 1.21, 1.45, 1.76, 1.80, and 2.40, respectively. Lower Ca/Si values indicate that the mixture has more CSH and a denser microstructure with increased mechanical strength. It has been reported in the literature that the compressive strength of cementitious materials increases with decreasing Ca/Si [69].

## 4. Conclusions

This research evaluated the mechanical and durability performances of engineered cementitious composites fabricated from various mineral additions. The following conclusions can be drawn based on the findings:Using mineral admixtures such as RHA, WHA, and GP can successfully improve the mechanical performance of ECC-based mortars. RHA and WHA showed greater enhancement in mechanical strength due to their finer particles. This improvement in mechanical strength is attributed to the pozzolanic reaction and the micro filler effect of mineral admixtures. The micro-filler effect distributes the hydration products more homogeneously in the available space, resulting in a much denser matrix. Moreover, the inclusion of PP fibers improves the mechanical properties of ECC.Water absorption, apparent porosity, and sorptivity were much lower for RHA, WHA than GP and control samples. These mineral admixtures increase the density of ECC-based mortars and improve the pore structure of the matrix by reducing the average pore size and sorptivity coefficient of composite microstructure.XRD, FTIR, SEM, and EDX results are consistent with mechanical and physical performances. Materials characterization showed that using an optimum amount of mineral admixtures imparts a dense and compact microstructure, reducing the ingress of chemicals, reducing permeability, ensuring water tightness and micro-cracks, and improving mechanical strength.Chemical durability results showed that RHA-based ECC specimens underwent lesser weight and compressive strength losses than other ECC mixtures when exposed to HCl and Na_2_SO_4_ solutions for 180 days. The changes in the compressive strength of ECC-based mortar exposed to acid and sulfate environments are primarily governed by the formation of expansive gypsum and salts. The porous medium governs the degradation by regulating the dissemination of ionic species of the pore solution towards the aggressive environment.UPV, ER, and RCPT performances of ECC mortars were enhanced due to mineral additions. SiO_2_ in RHA, WHA, and GP composition combined with calcium hydroxide to form additional CSH. The microstructure of ECC is affected by this reaction, which reduces the interpore connectivity, resulting in a dense and compact microstructure.

## Figures and Tables

**Figure 1 materials-15-02880-f001:**
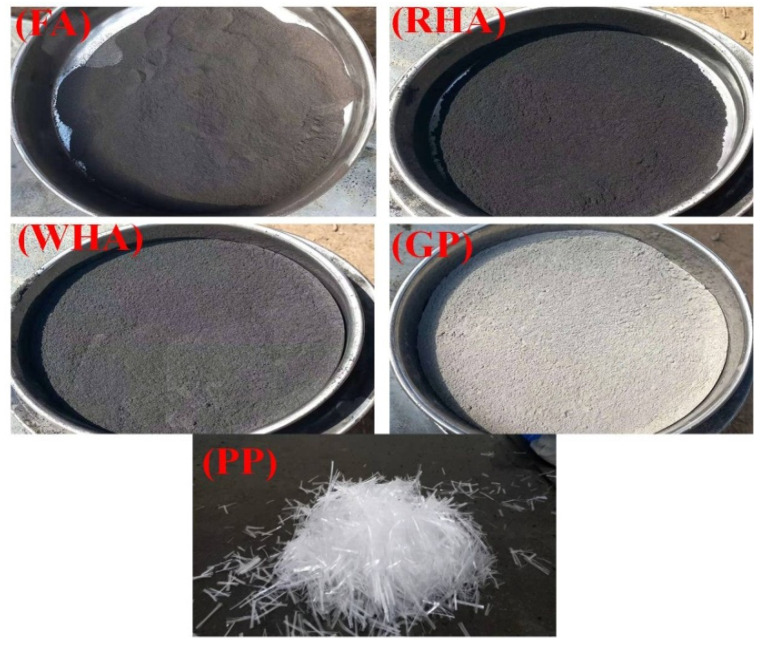
The visual aspect of mineral admixtures and the PP fibers used in the current study.

**Figure 2 materials-15-02880-f002:**
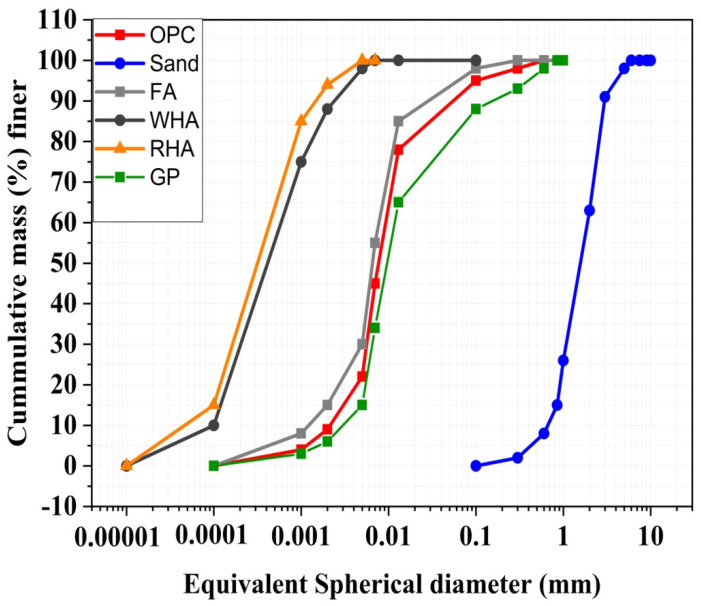
Particle size distributions of cement, sand, and different mineral admixtures.

**Figure 3 materials-15-02880-f003:**
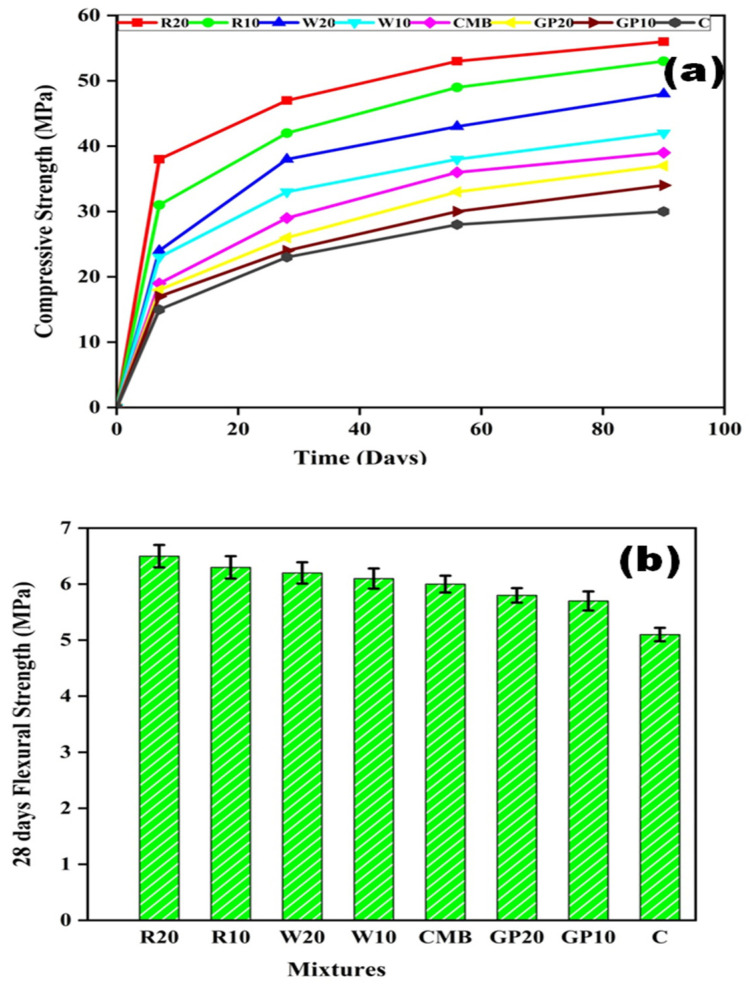
Strength of ECC specimens, (**a**) compressive strength (MPa), (**b**) flexural strength (MPa). Here, error bars represent the variability of flexure strength results.

**Figure 4 materials-15-02880-f004:**
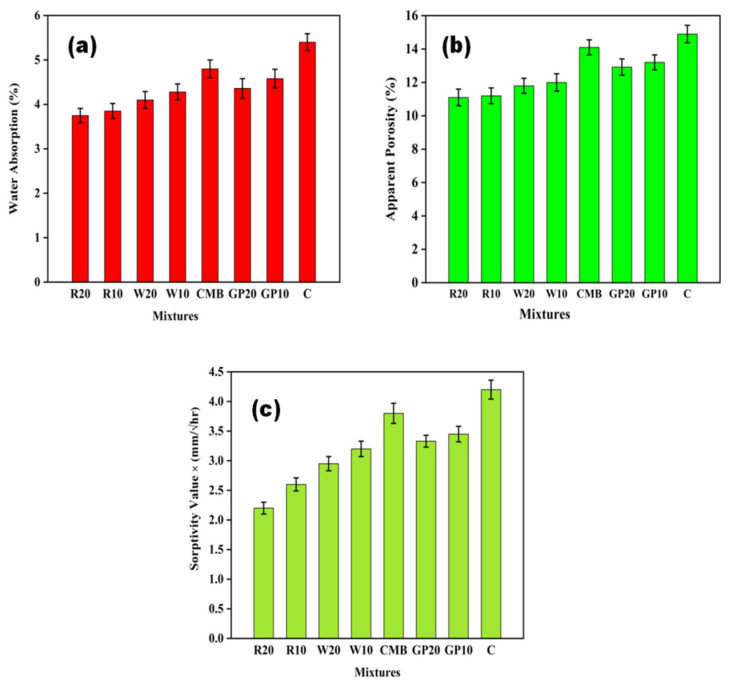
(**a**) water absorption (%), (**b**) apparent porosity (%), and (**c**) sorptivity (mm/√hr) of ECC mixes after 28 days. Here, error bars represent the variability of results.

**Figure 5 materials-15-02880-f005:**
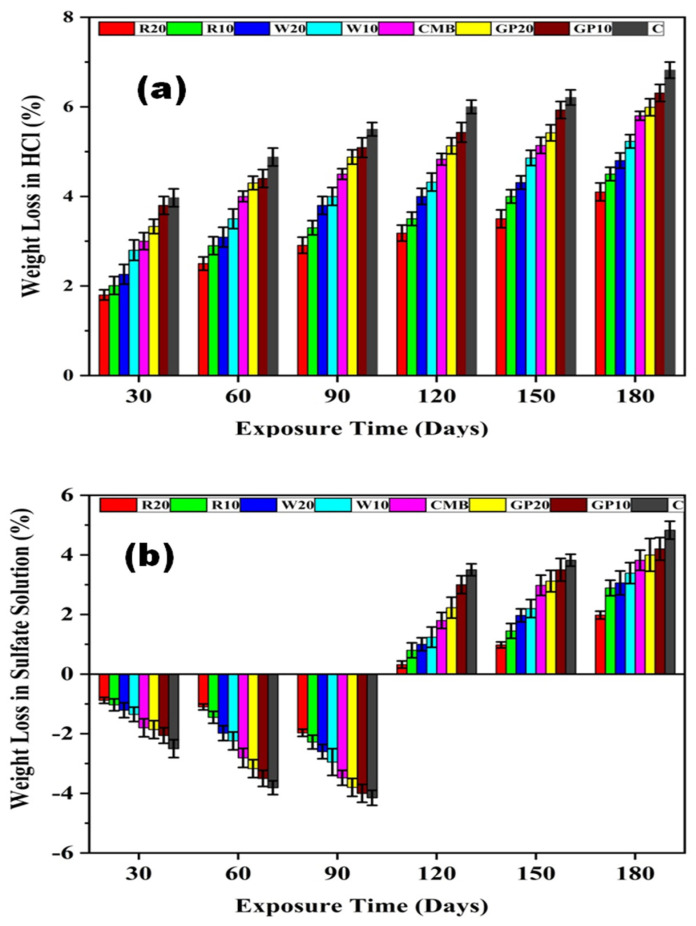
Weight losses (%) of all mix proportions of ECC in (**a**) an HCl solution and (**b**) sulfate solution. Here, error bars represent the variability of weight loss results.

**Figure 6 materials-15-02880-f006:**
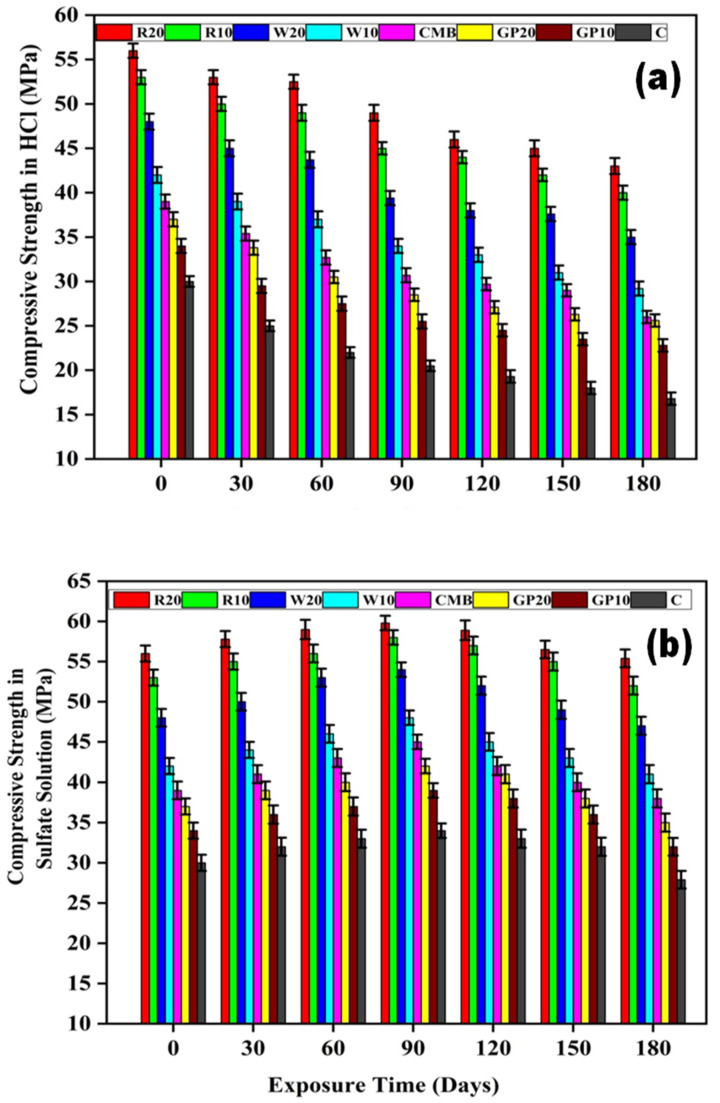
(**a**) compressive strength (MPa) of ECC mix proportions in HCl solution, (**b**) compressive strength (MPa) of ECC mix proportions in sulfate solution (MPa). Here, error bars represent the variability of residual compressive strength results.

**Figure 7 materials-15-02880-f007:**
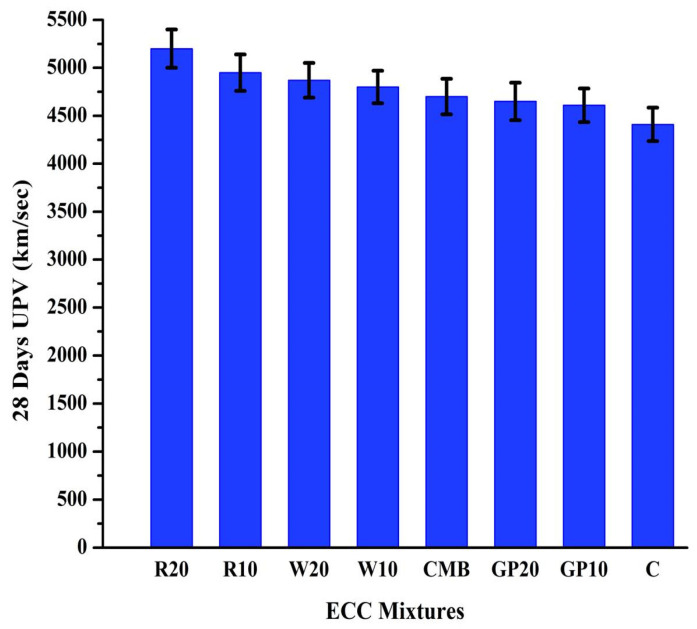
Ultrasonic pulse velocity test of ECC mixes after 28 days (km/s). Here, error bars represent the variability of UPV results.

**Figure 8 materials-15-02880-f008:**
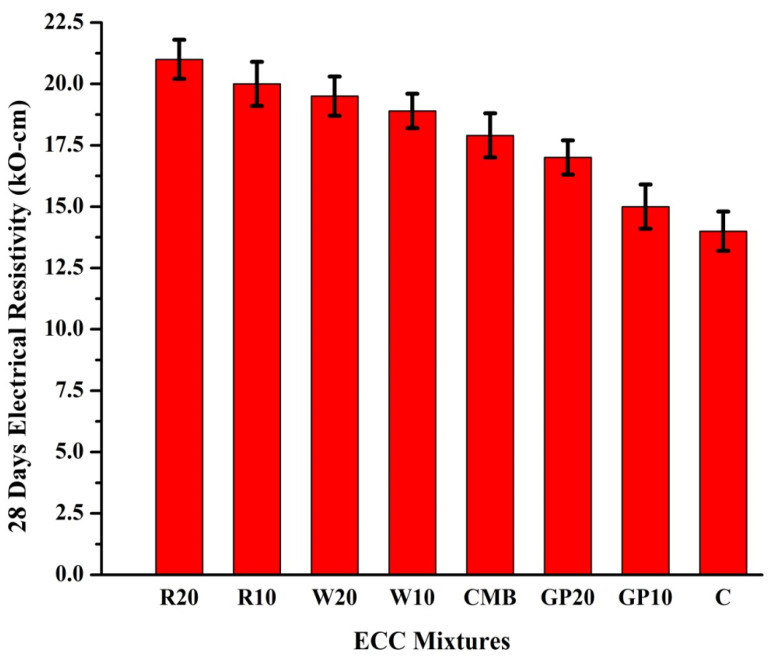
Electrical resistivity test results of ECC mixes after 28 days (kΩ-cm). Here, error bars represent the variability of ER results.

**Figure 9 materials-15-02880-f009:**
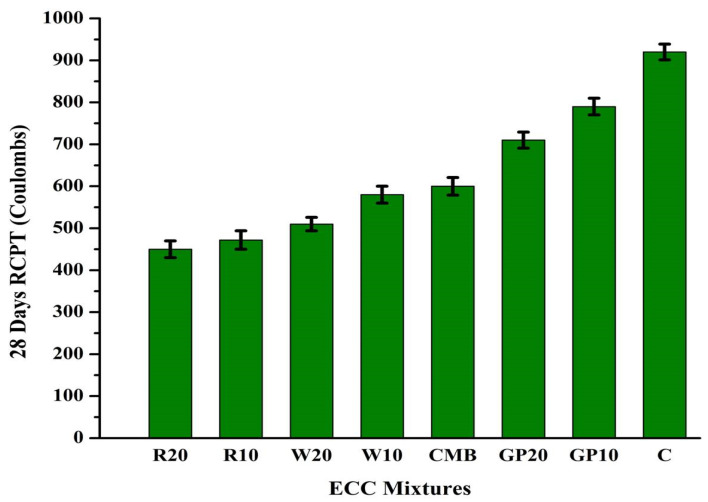
Rapid chloride penetration test of ECC mixes after 28 days (coulombs). Here, error bars represent the variability of RCPT values.

**Figure 10 materials-15-02880-f010:**
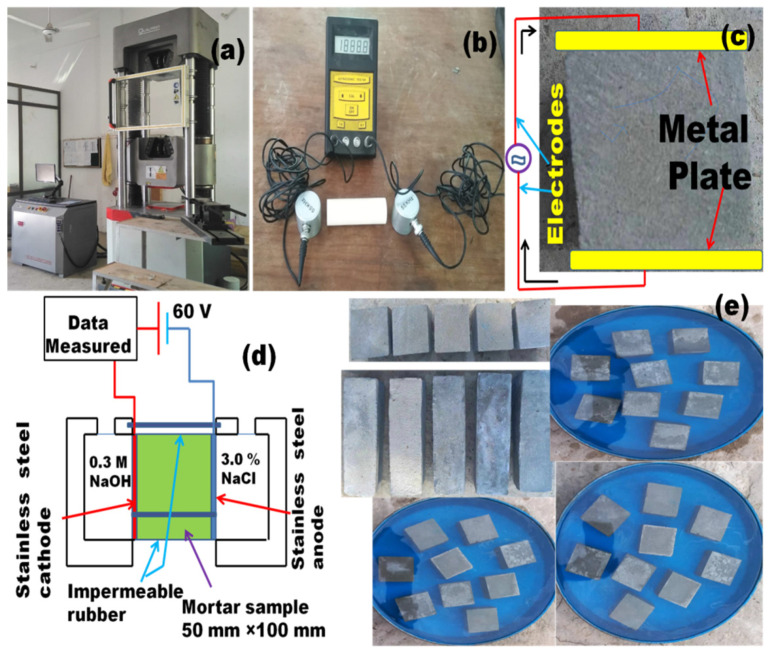
(**a**) mechanical testing setup; (**b**) UPV test device; (**c**) ER test schematic; (**d**) RCPT test setup; (**e**) specimens used for various durability related tests.

**Figure 11 materials-15-02880-f011:**
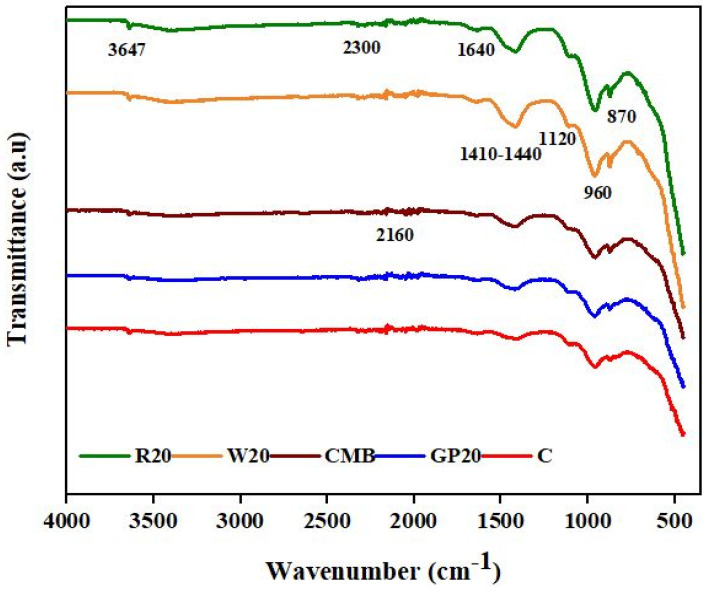
FTIR transmission spectra of control, GP20, CMB, W20, and R20 cured in water for 28 days.

**Figure 12 materials-15-02880-f012:**
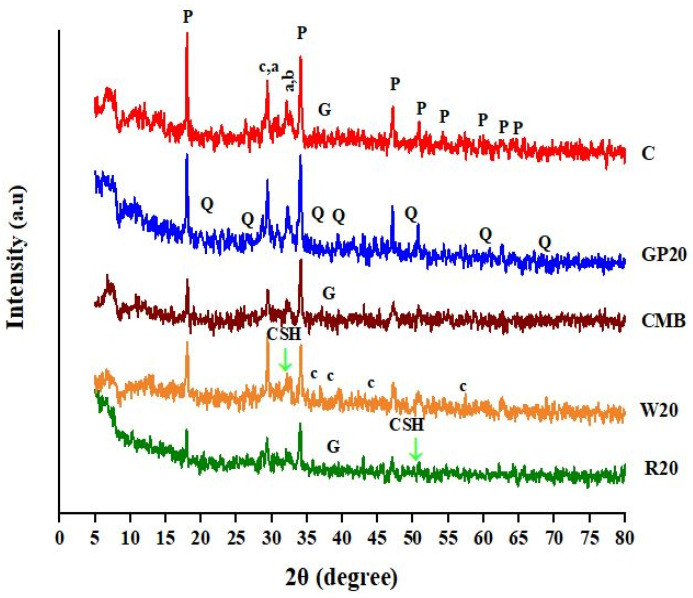
X-ray diffraction pattern of 28-day, hydrated samples of the control, GP20, CMB, W20, and R20. P: portlandite, Q: quartz, CSH: calcium-silicate hydrate, a: alite, b: belite, c: calcite; G: gypsum.

**Figure 13 materials-15-02880-f013:**
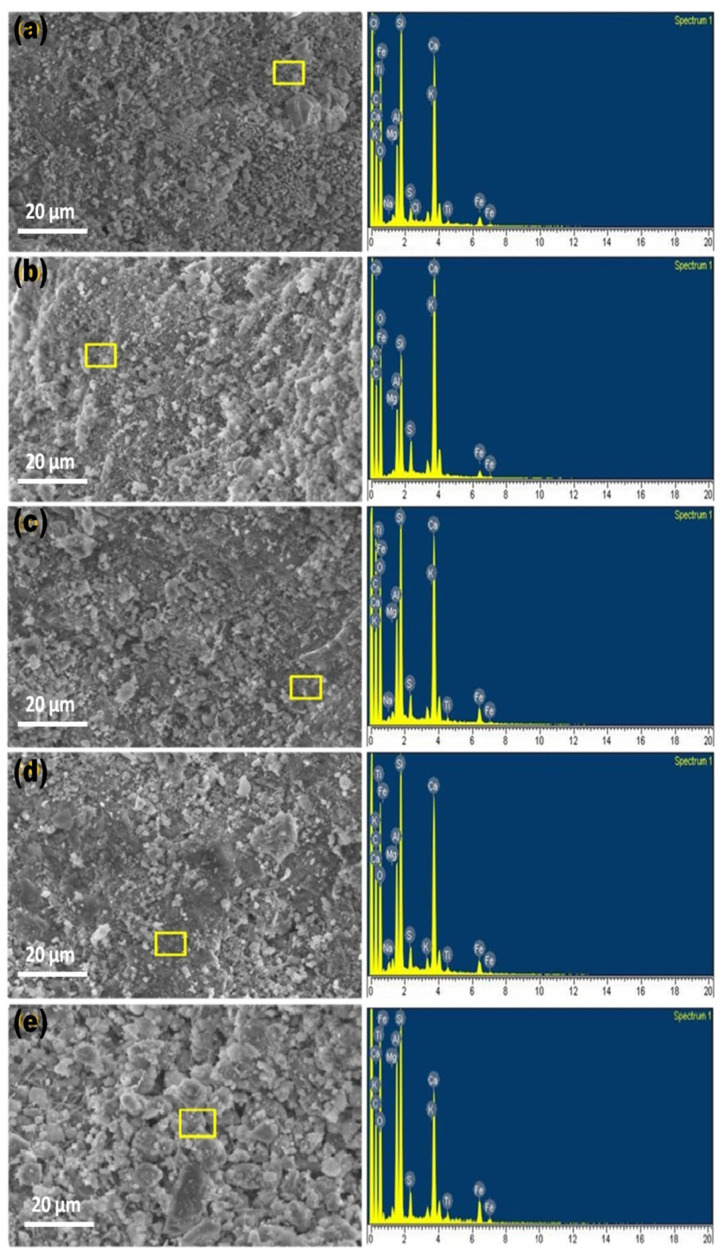
SEM and EDX images (**a**) R20, (**b**) W20, (**c**) CMB, (**d**) GP20, and (**e**) C of ECC mix proportions, which are cured in water for a period of 28 days.

**Table 1 materials-15-02880-t001:** Chemical compositions of mineral admixtures (OPC (ordinary Portland cement), FA, RHA, WHA, and GP).

Chemical Composition (%)	OPC	FA	RHA	WHA	GP
SiO_2_	20.78	58	93.2	72.7	68.1
Fe_2_O_3_	2.99	3.58	0.1	-	0.4
Al_2_O_3_	5.81	29.08	0.4	0.48	2.0
CaO	62.18	3.6	1.1	10.6	13.6
SO_3_	1.89	1.8	0.9	6.13	-
MgO	1.52	1.91	0.1	2.20	1.3
TiO_2_	-	1.75	-	-	-
Na_2_O	-	2.00	0.1	5.41	7.2
P_2_O_5_	-	-	-	4.68	-
K_2_O	-	0.73	1.3	11.4	0.6
Free Lime	0.68	-	-	-	-
IR	0.3	-	-	-	-
LOI	2.00	2.00	3.7	5.5	0.49

**Table 2 materials-15-02880-t002:** Demonstrating the properties of fibrillated polypropylene (PP) fibers.

Properties	Value
Length (mm)	13
Diameter (µm)	32
Tensile Strength (MPa)	520
Elastic Modulus (GPa)	4.2
Elongation (%)	4

**Table 3 materials-15-02880-t003:** Representing the composition of ECC mixtures (kg/m^3^).

Mix ID	Controlled (C)	RHA 10% (R10)	RHA 20% (R20)	WHA 10% (W10)	WHA 20% (W20)	GP 10% (GP10)	GP 20% (GP20)	Combined 20% (CMB)
OPC	381.6	381.6	381.6	381.6	381.6	381.6	381.6	381.6
FA	890.4	763.2	636.0	763.2	636.0	763.2	636.0	636.0
RHA	0	127.2	254.4	0	0	0	0	84.8
WHA	0	0	0	127.2	254.4	0	0	84.8
GP	0	0	0	0	0	127.2	254.4	84.8
Sand	462	462	462	462	462	462	462	462
Water	318	318	318	318	318	318	318	318
HRWR	15.3	15.3	15.3	15.3	15.3	15.3	15.3	15.3
Fiber	26	26	26	26	26	26	26	26

**Table 4 materials-15-02880-t004:** Wt. % of various elements of all tested ECC mixes.

Sample Code	Hydration Time (Days)	Wt. % of Elements Measured from EDX Analysis											
		C	O	Na	Mg	Al	Si	S	K	Ca	Ti	Fe	Cl
CMB	28	34.56	30.97	0.20	0.31	3.53	8.50	1.28	0.93	15.00	0.49	4.15	-
R20	28	28.59	35.96	0.10	0.31	3.28	9.72	1.08	1.03	11.80	0.39	2.97	0.18
W20	28	24.74	34.88	-	0.24	3.23	9.10	2.25	1.37	15.00	-	2.56	-
GP20	28	30.94	33.17	0.47	0.48	4.04	8.40	1.28	0.83	15.20	0.54	4.27	-
C	28	26.45	35.06	-	0.53	6.18	7.53	1.53	0.76	18.10	0.82	7.61	-

## Data Availability

The authors declare that all data supporting the findings of this study are available within the article.

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
