# Peer review of "Effects of Various Mineral Admixtures and Fibrillated Polypropylene Fibers on the Properties of Engineered Cementitious Composite (ECC) Based Mortars"

_materials, 2022, doi:10.3390/ma15082880_

Round 1

Reviewer 1 Report

The paper is very well structured and is suitable for the scope of the journal and the following must be addressed before it is published: 

1) The conclusion need to be amended by presenting the aim of the study then listing the main findings. 

Author Response

The authors are grateful to the editor and kind reviewers for their valuable input/comments in improving the quality of the manuscript. The authors have revised the manuscript as advised by the reviewers.

The point-by-point response to the reviewer's comments can be found in the uploaded file.

Thank you

Dr. Bakht Zamin

Reviewer 2 Report

The article contains interesting research on the use of alternative fly ash substitutes in the production of mortars in accordance with sustainable development. Many properties of the mortars have been tested and sufficient analyzes have been made. However, in order for the manuscript to be considered for publication, its quality should be improved. Below are my comments:

Introduction:

The introduction was written correctly. Defines the materials used in the research and, on the basis of the literature review, presents the influence of the admixtures used in the work on the properties of mortars and concretes.

  • Line 122-123: “According to the Paris agreement, fly ash production is limited, so looking to other materials”
  • Line 129: “ordinary concrete” or mortar?

Materials and methods:

  • In chapter 2.1. the origin and process of preparation of wheat husks are described, while rice husks and glass powder are not described. Please complete this information.
  • Table 1: Subscripts should be used in the chemical formulas of oxides.
  • In caption Figure 1: must the word "Minerals" start with a capital letter?
  • Line 160: „and the same amount of HRWR, i.e., 1.2% 160 by weight” - 1.2% by weight of cement? or cement + ash? or 1.2% of the total weight of the mixture? This should be clarified.
  • Line 161-163: At which stage was sand added?
  • Table 3: (kg/m3): Superscript should be used
  • Line 166: Superplasticizer is marked with the symbol (SP), while in other parts of the text as (HRWR). The HRWR symbol should be explained.
  • Line 177: “The mixture was cast and tamped..”. How were the samples compacted?
  • Line 188: 40×40×160 mm3: Superscript should be used
  • Line 195: The full stop (.) is missing at the end of the sentence.
  • Line 201: “concrete surface” or mortar surface?
  • Line 204: The formula should be numbered e.g. (1), and symbols must have units in the legend.
  • 2.3.3. Sorptivtity: What sample dimensions were used and how many samples from the recipe?
  • 2.3.4. Ultrasonic pulse velocity (UPV), Electric resistivity (ER), Rapid chloride penetration test (RCPT): A photo of the test stand would be helpful
  • Line 217: Formula (1) should be corrected. "ER" was written twice. Symbols must have units.
  • Line 222: “NaC1” - should be "l" instead of "1"
  • Line 229: “Sulfate” shouldn't be lowercase?
  • 2.3.5. Acid and sulfate resistance: What sample dimensions were used and how many samples from the recipe?
  • Overall, this chapter (Materials and methods) lacks photos of the samples and the test stand. Selected photos would be helpful

Results and discussion:

  • Figure 3: What do the error bars mean? This should be explained in the figure caption. Above the bars it would be helpful to put the average strength values. In line graphs (Fig.3a) it will be difficult to insert error bars, but it is worth writing something about the deviations in the results in the text.
  • Figure 4: What do the error bars mean? This should be explained in the figure caption. Above the bars it would be helpful to put the average values. This will be more legible than inserting the results in the text (line 290-292)
  • Figure 5, 6: What do the error bars mean? This should be explained in the figure caption.
  • 3.5. Compressive strength analysis: I believe that the title of the subchapter should be modified eg to "Changes in compressive strength after exposure to chemical aggression" or otherwise. In the current version, the title seems to be repeating itself as the compressive strength has already been analyzed above.
  • Figure 7 and 8 and 9: What do the error bars mean? This should be explained in the figure caption. Above the bars it would be helpful to put the average values.
  • Each Figure caption includes a unit, e.g. (%), (km / sec). However, figure 3 caption does not include the unit (MPa)
  • Figure 12: Please provide a magnification of the photos.
  • In Fig 12 it is written that these are photos of samples matured for 28 days, and in Table 4 it is written that the hydration period was 30 days.

The list of literature should be formatted in accordance with the requirements of the publishing house.

Author Response

(The authors gave the same response as above.)

Reviewer 3 Report

The manuscript presented gives an almost full idea of how the addition of natural puzzolans together with fly ash and cement, improve the durability of cement-based mortars. The following are some concerns that the authors should take into consideration to improve the manuscript:

  1. The main objective was to obtain possible improvements on the physical properties of mortars if cement, FA, and fibers are included in this material. Addition of RHA, WHA, and GP are based on substitution of FA, maintaining cement, fiber content, water, and sand amounts constant. What was the flow properties of all of these mortars tested? Authors should include in Table 3 the flow according to ASTM C-1437.
  2. Why did the authors not prepared a Control mortar without the addition of FA and fibers to see improvements from a typical cement-based mortar material?
  3. There is a mistake in equation 1: ER is repeated.
  4. There is a mistake in the definition of Wenner method to obtain resistivity. Wenner method uses 4-point test procedure, and the authors apparently used the direct method, according to equation 1 presented. Please, verify and do the needed changes.
  5. Why the values of UPV are only for 28-day age? Why was not obtained at 90-day age?
  6. There are no correlation graphs where the durability indices obtained are compared between each other, as previous authors are presented in the literature. For example, correlation between 1) ER and fc, 2) ER vs Sorptivity; 3) fc vs flexure strength; 4) ER vs UPV; 5) ER vs RCPT. Is it possible to include them to see some tendencies?

Author Response

The authors are grateful to the editor and kind reviewers for their valuable input/comments in improving the quality of the manuscript. The authors have revised the manuscript as advised by the reviewers.

Kind Regards

Dr. Bakht Zamin
